# Respiratory Syncytial Virus (RSV) Disease and Prevention Products: Knowledge, Attitudes, and Preferences of Kenyan Healthcare Workers in Two Counties in 2021

**DOI:** 10.3390/vaccines11061055

**Published:** 2023-06-02

**Authors:** Bryan O. Nyawanda, Victor A. Opere, Joyce U. Nyiro, Elisabeth Vodicka, Jessica A. Fleming, Ranju Baral, Sadaf Khan, Clint Pecenka, Jorim O. Ayugi, Raphael Atito, James Ougo, Godfrey Bigogo, Gideon O. Emukule, Nancy A. Otieno, Patrick K. Munywoki

**Affiliations:** 1Kenya Medical Research Institute, Center for Global Health Research, Kisumu P.O. Box 1578-40100, Kenya; 2Kenya Medical Research Institute-Wellcome Trust Research Programme, Kilifi P.O. Box 230-80108, Kenya; 3Program for Appropriate Technology in Health—PATH, Seattle, WA 98121, USA; 4US Centers for Disease Control and Prevention, Nairobi P.O. Box 606-00621, Kenya

**Keywords:** attitudes, healthcare workers, knowledge, preferences, prevention products, respiratory syncytial virus

## Abstract

Respiratory syncytial virus (RSV) is a major cause of lower respiratory tract infection (LRTI) among infants under 6 months of age. Yet, in Kenya, little is known about healthcare workers’ (HCWs) knowledge, attitudes, and perceptions around RSV disease and the prevention products under development. Between September and October 2021, we conducted a mixed methods cross-sectional survey to assess HCWs’ knowledge, attitudes, and perceptions of RSV disease and RSV vaccinations in two counties. We enrolled HCWs delivering services directly at maternal and child health (MCH) departments in selected health facilities (frontline HCWs) and health management officers (HMOs). Of the 106 respondents, 94 (88.7%) were frontline HCWs, while 12 were HMOs. Two of the HMOs were members of the Kenya National Immunization Technical Advisory Group (KENITAG). Of the 104 non-KENITAG HCWs, only 41 (39.4%) had heard about RSV disease, and 38/41 (92.7%) felt that pregnant women should be vaccinated against RSV. Most participants would recommend a single-dose vaccine schedule (n = 62, 58.5%) for maximal adherence and compliance (n = 38/62, 61.3%), single dose/device vaccines (n = 50/86, 58.1%) to prevent wastage and contamination, and maternal vaccination through antenatal care clinics (n = 53, 50%). We found the need for increased knowledge about RSV disease and prevention among Kenyan HCWs.

## 1. Introduction

Respiratory syncytial virus (RSV) remains a major cause of lower respiratory tract infection (LRTI) in children worldwide, and infants under 6 months of age bear the greatest burden [1,2,3]. Currently, there are no licensed RSV vaccines targeting this age group, and the only pharmaceutical intervention available is a short-acting monoclonal antibody (mAb), Palivizumab^®^, with a modest efficacy of 50% at reducing RSV-associated hospitalizations [4]. Due to its prohibitive cost and need for multiple rounds of dosing during RSV season, Palivizumab^®^ is rarely used in Kenya and other low- and middle-income countries (LMICs) [4]. However, there has been significant progress over the past decade in the development of long-acting mAbs and maternal RSV vaccines [5,6]. Nirsevimab^®^ (a long-acting mAb, NCT03979313) [7] was approved by the European Union for use on the 4th of November 2022, while a phase IIb maternal vaccine (NCT04032093) [8] is very close to licensure after the US. Food and Drug Administration (FDA) granted it a Breakthrough Therapy designation in March 2022 [9]. Several other vaccine and mAb formulations are in varying stages of clinical development and approvals [5,6].

As these RSV prevention products targeting pregnant women (maternal vaccines) and newborns (long-acting mAbs) become available for introduction in the market, establishing the knowledge, perceptions, and acceptance of various stakeholders is crucial. Previous studies pointed out that vaccine safety, strong national recommendations, and healthcare provider recommendations are top facilitators for maternal vaccine acceptance [10,11]. Moreover, studies exploring maternal vaccine acceptance in Kenya observed that healthcare workers (HCWs) view themselves as the responsible persons for vaccine promotion and that pregnant women would accept vaccines if recommended by HCWs [12,13,14,15].

Despite the high burden of RSV disease among young children, its epidemiology is not well known to clinicians worldwide. Only two previous studies from developed countries have been published, and they document little knowledge of RSV disease and the use of the existing immunoprophylaxis (palivizumab) [16,17]. Furthermore, no studies have explored the potential vaccine acceptance for RSV immunization in sub-Saharan Africa. This study aimed to assess the current knowledge, attitudes, and perceptions around RSV disease and RSV prevention products in development among the HCWs serving in various health facilities in two diverse counties in Kenya. It also explored the HCWs’ preferences regarding product delivery to gain insights into vaccine product characteristics that would inform a feasible and sustainable introduction and implementation in low-resourced countries. Taken together, the presented data provide useful information to guide the introduction of RSV prevention products in Kenya and other LMICs. 

## 2. Methods

### 2.1. Study Design, Sites, and Population

A mixed methods cross-sectional study was conducted between 13 September and 5 October 2021 on frontline HCWs (defined as HCWs delivering services directly to mothers at the maternal and child health (MCH) departments) in five health facilities in Siaya County (Siaya County Referral Hospital, Bondo Sub-County Hospital, Bama Hospital, St. Elizabeth Lwak Mission Hospital, and Ong’ielo Health Centre) and three health facilities in Nairobi County (Mbagathi Hospital, Mama Lucy Kibaki Hospital, and Tabitha Medical Clinic) in Kenya. Bama Hospital, St. Elizabeth Lwak Mission Hospital, and Tabitha Medical Clinic are private hospitals while the rest are public hospitals. In the selected health facilities, all frontline HCWs with a regular assignment to the maternal and child health (MCH) departments were approached to participate. Additionally, health management officers (HMOs) in the two counties that were in charge of vaccine distribution, policy, and management and members of the Kenya National Immunization Technical Advisory Group (KENITAG) were purposively selected and interviewed. This study was a follow up to a larger cross-sectional survey examining the factors that shape the acceptance of maternal vaccines in Kenya, which was a collaboration between the Kenya Medical Research Institute (KEMRI); the US Centers for Disease Control and Prevention (CDC) in Kenya; and Emory University in Atlanta, GA, USA [13,14].

### 2.2. Data Collection and Questionnaire

Trained research assistants approached, obtained consent from, and conducted in-depth interviews with HCWs in the selected health facilities and HMOs by using a semistructured questionnaire in the respondent’s language of choice (either English, Kiswahili, or Luo). The interviews were rescheduled upon the respondents’ request. The questionnaire included quantitative and qualitative questions on knowledge, attitudes, and perceptions around RSV disease and RSV prevention products. Additional questions on HCWs’ preferences for the number of doses per vaccine vial (i.e., vaccine vial sizes), scheduling, and immunization strategies were included. The questionnaire was pilot tested at Siaya County Referral Hospital and necessary adjustments were made to ensure reliability. Knowledge of RSV burden and prevention products were only assessed among non-KENITAG HCWs. However, in-depth interviews were conducted with members of the KENITAG regarding the requirements for the successful introduction of RSV vaccines in Kenya. 

### 2.3. Data Analysis

The quantitative data were analyzed with Stata version 16 (Statacorp, College Station, TX, USA), while the qualitative data were analyzed by using Nvivo version 11 Pro (QSR International Pty Ltd., Victoria, Australia). Frequencies and percentages were used to summarize the categorical variables for the close-ended questions. Chi-square and Fisher’s exact tests were used to compare the differences in knowledge by group (HCWs vs. HMOs) and by counties. Theme codes were used to summarize the open-ended (qualitative) questions, especially the respondents’ attitudes and preferences. The lead authors (BON, VAO), assisted by RAs, familiarized themselves with all the transcripts before entering the data into Nvivo. Abstracted sections were assigned codes as the authors compared key points by consensus. 

### 2.4. Ethical Considerations

Ethical clearance for the study was obtained from KEMRI’s scientific and ethics review unit (SERU) (SERU # 3292) and Emory University’s institutional review board (IRB) (IRB00089673), with CDC reliance on non-CDC IRB (CDC Protocol #6974). Written informed consent was obtained from all participants.

## 3. Results

### 3.1. Participant Characteristics

A total of 106 HCWs were approached and interviewed (100% response rate). Of the 106 respondents, 94 (88.7%) were frontline HCWs with a regular assignment to the MCH clinics (54 from Siaya and 40 from Nairobi) and 12 were HMOs (6 in each county: Siaya and Nairobi) (Table 1). Of the 106 HCWs, 64 (60.4%) were female, 65 (61.3%) were nurses, 67 (63.2%) had diploma certificates, and 49 (46.2%) had 5–10 years of experience (Table 1). More than half of the frontline HCWs were female (n = 59, 62.8%), nurses (n = 65, 69.2%) with diploma-level training (n = 66, 70.2%), and 45 (47.9%) had 5–10 years of experience. A majority (9/12) of the HMOs had a bachelor’s degree, 8 of 12 had over 15 years of experience, and two were members of the Kenya National Immunization Technical Advisory Group (KENITAG).

### 3.2. Knowledge and Perceptions on RSV-Associated Disease and RSV Vaccination

Of the 104 non-KENITAG HCWs, only 41 (39.4%) had heard about RSV disease, and of those, most highlighted cough, difficulty breathing, fever, and runny nose as some of the symptoms. Only 2/104 (1.9%) were aware of RSV prevention products (Table 2). Nearly all (n = 39/41, 95.1%) HCWs who had heard about RSV disease were unaware of RSV prevention products in the market or in development but would recommend the vaccine if available. The vast majority (38/41, 92.7%) felt that pregnant women should be vaccinated against RSV. These participants highlighted the factors to be considered before recommending the RSV vaccine to pregnant women, which included vaccine efficacy, safety of the vaccine, presence of underlying conditions among pregnant women, age, signs and symptoms for active disease, trimester of administration, vaccine side effects, and the duration of protection to the mother and the baby. The participants also highlighted counseling, health education, stating benefits to the patient, evidence of better outcomes among the vaccinated, and creating awareness in the health workers as additional steps that would be helpful in encouraging the uptake of an effective maternal RSV vaccine among pregnant women. The two members of the KENITAG stated that data on the burden of RSV, vaccine safety, efficacy, and cost effectiveness would be necessary to recommend the introduction of RSV vaccines and monoclonal antibodies. There was no difference in the proportions of HCWs who had heard of RSV disease between the two counties: 39.4% in Siaya vs. 41.7% in Nairobi, *p* = 0.58 (Appendix A).

Half of the HCWs (n = 51, 48.1%) indicated that the addition of the new services required to support the introduction of RSV vaccination (e.g., diagnostic, therapeutic, counseling) would interrupt antenatal care (ANC) services due to an increased workload, additional responsibilities, and the competition of tasks. It would be necessary to add new personnel and resources to accommodate the additional services. Other HCWs (n = 17, 16.0%) also suggested that the addition of a new service may reduce the ANC uptake. There were concerns about mothers going unattended because of longer waiting periods, which would affect the smooth running of ANC services as more time would be required with a single client. Some participants (n = 8, 7.7%) also stated that some pregnant women were likely to delay ANC attendance as they would stay back to observe the vaccination outcomes in other women.

One frontline HCW stated:


*“…the main issue is the competition of tasks now, because that [vaccination] is an additional task. Just competition of tasks, lack of resources for the same task and even sometimes just lack of motivation because now you are used to doing something one way then you are being added more responsibilities without commensurate remuneration.”*


Yet another frontline HCW said:


*“… Impact one, it will increase workload again because you need staff to administer the same [vaccine]. So, two, again, depending on the mother’s understanding, some of them may not come for the ANC visit. They will want to wait to see what happens to so and so when vaccinated, yes there will be, [even] if it is not much but, a little reduction in the ANC attendance…”*


However, some participants (n = 16, 15.1%) felt that the addition of new services would not have a significant impact on the uptake and continuity of ANC services. This view was shared by the two KENITAG members that when available, the vaccination should be integrated with existing services to form a package, and sensitization should be conducted as appropriate. The respondents shared that it would have a positive effect if manpower was increased and finances were provided. One HMO stated:


*“… I don’t think it has such a big effect. Because if it’s a new introduction and you have been trained about it and you know what you need to do about it, maybe the first time it will bring some delay before you catch up, but after that I think everything will be okay. The first few days there are of course some hiccups here and there and of course people know how things are done …”*


### 3.3. Awareness and Acceptance of Maternal Vaccines 

In general, nearly all (105/106, 99.0%) HCWs were aware of the vaccines currently available for pregnant women in Kenya, with tetanus toxoid being the most popular vaccine (104, 99.0%) (Table 3). When asked which maternal immunizations they would recommend to pregnant women, most of the participants stated that they would recommend the tetanus toxoid vaccine (n = 80, 76.2%) followed by the COVID-19 vaccine (n = 39, 37.1%) during pregnancy. HMOs were significantly more likely to recommend COVID-19 vaccination for pregnant women compared to frontline HCWs (66.7% vs. 33.3%, *p*-value = 0.03) (Table 3). Furthermore, HCWs from Nairobi County were more likely to recommend COVID-19 vaccination for pregnant women than Siaya HCWs (27.5% vs. 7.5%, *p*-value < 0.01) (Appendix A). Twenty-two (21.0%) of the HCWs, with 20 from Siaya County, would recommend malaria vaccination for pregnant women if available (*p*-value < 0.01). When asked about the rationale for their recommendations, most of the HCWs would recommend maternal immunization to protect the mother (n = 93, 88.6%) and the baby (n = 78, 74.3%). In terms of the general timing for vaccinating pregnant women, most HCWs (n = 63, 59.4%) perceived that vaccination at 16–32 weeks gestation would result in the optimal effectiveness of the vaccine. Notably, most participants (n = 92, 86.8%) were unaware of alternative prevention products such as monoclonal antibodies.

### 3.4. Healthcare Workers’ Preferences for Vaccine Vial Sizes and Scheduling

Three quarters (n = 79, 74.5%) of the participants indicated that most pregnant women would prefer a single-dose vaccine schedule. A slightly lower number of HCWs (n = 62, 58.5%) would recommend single-dose vaccine schedules (Table 3). Of the 62, 38 (61.3%) would recommend single-dose vaccine schedules for maximal adherence and compliance by pregnant women. Other reasons included the reduction in the number of injections and side effects (n = 25/62, (40.3%)) and the reduction in workload for HCWs (n = 6/62, (9.7%)). One frontline HCW stated:


*“… Because it will be administered once during the period the mom is carrying the pregnancy. She will not be coming to the clinic again, and you know for adherence for pregnant women especially in rural areas, coming to the clinic is very … is usually a challenge when you look at logistics. And also, for the registers, it will reduce the workload, especially for the nurses at the MCH. So I would prefer a single dose.”*


All the participants who would recommend multidose vaccine schedules (n = 44/106, 41.5%) reported maximizing the effectiveness for both the mother and their newborns as the main reason.

Regarding vaccine vial size preference, only 86/106 HCWs were available for a follow-up interviews. More HCWs (n = 50/86, 58.1%) would recommend a single-dose/device vaccine mainly to prevent wastage and contamination. On the other hand, 23 out of the 26 (88.5%) frontline HCWs and 10 HMOs who preferred multidose vials were mainly concerned about vaccine storage and saving of operation costs. In addition, more than half of the participants (n = 57, 53.4%) indicated that they were concerned about adverse birth outcomes following maternal immunization, which included birth complications, the premature rupture of membranes, reactions to the vaccine, deformities, abortion, side effects, teratogenicity, and mental and growth retardation.

### 3.5. Immunization Strategies, Constraints, and Tools to Motivate Pregnant Women to Be Vaccinated

Half of the respondents (n = 53, 50%) stated that for the purpose of sustainability, incorporating the vaccine into the existing ANC platform would be the most feasible and most effective maternal vaccination strategy compared to one-time mass campaigns (Table 3). One KENITAG member stated:


*“…It could be multi-faceted although for sustainability it could be through ANC; through ANC, it is sustainable because these other ones [campaigns and mobile outreaches] are too expensive. I know even studies that are thinking of doing the routine [immunization] and probably just come once in a while to do the campaigns, but the campaigns are too expensive…”*


Some of the constraints noted by the participants about counseling pregnant mothers on the importance of vaccination included passing inadequate information due to time pressures that arise when women arrive at the clinic late, language barriers, and illiteracy. The HCWs also mentioned that client load and low staffing makes it difficult to cover all areas during the counseling sessions. A lack of adequate training for providers and a lack of infrastructure were also mentioned as barriers to proper counseling. One HCW stated: 


*“…those who come late they will not get any information, and by that time when they are coming in, maybe you are busy, and you are looking at the queue and you want to clear it. You will not give the whole information to that mother …”*


While another observed that language could be a challenge:


*“…Maybe there can be language barrier, when you are talking to the mother, and [she] lacks understanding of the language being used…”*


The participants also shared that counseling using posters in the ANC clinic rooms could help in assuring and convincing pregnant women to receive vaccines. They also mentioned that word of mouth by clients (i.e., pregnant women sharing amongst themselves); the engagement of mentor mothers (older mothers who advise relatively younger mothers); the use of community health volunteers (CHVs) within their communities; the use of advertisement media such as the newspaper, television, and posters; health education; community mobilization and outreaches; home visits; and sensitization through administrative officers and the church would be appropriate sensitization strategies to be considered to increase vaccine acceptance. Three HCWs stated that:


*“…There is door-to-door, there is linking up with the CHVs—the community health volunteers because they… within their community they know that as at now I’m having five pregnant women and they know where they can be able to find them. So door-to-door and also CHVs”*



*“…health education and then one is to use some focal people in the ground like CHVs. Once these CHVs have been empowered with knowledge, they are in a good position of convincing these mothers…”*



*“…I think one of the things you need to do is to do a lot of advocacy at the community level of the importance of ANC services earlier…”*


## 4. Discussion

A low awareness of RSV disease (39%) and RSV prevention products either in the market or in development (2%) among Kenyan HCWs in the two counties of Siaya and Nairobi was observed despite RSV being the leading cause of viral pneumonia in LMICs [2], which is a substantial burden in Kenya [18]. These findings further underscore the need for more RSV advocacy and information sharing through training and continuous medical education (CMEs) among frontline HCWs in Kenya. For future maternal RSV prevention products, close to two-thirds of the HCWs would prefer a single-dose vaccine schedule and single-dose vial/device formulation to optimize adherence and minimize vaccine wastage and contamination. Vaccine manufacturers and policymakers need to consider these HCW preferences as new RSV prevention products are developed and released to the markets to ensure successful uptake among end users.

Studies from high-income countries (HICs) have also reported unsatisfactory or low knowledge levels of RSV epidemiology and prevention among HCWs [16,17,19]. To the best of our knowledge, there are no published studies on RSV awareness in sub-Saharan Africa. The findings in this study and those from HICs highlight a glaring knowledge gap of RSV that exists among HCWs despite its significant global disease burden [1], which necessitates the prioritization of creating awareness, possibly through continuous medical education, seminars, and conferences. Key issues to be addressed should include RSV clinical presentation, disease epidemiology, burden, and available or emerging prevention products. The awareness creation efforts should also extend to the general population given the high burden of disease among the young children and the eminent licensure and introduction of RSV prevention products such as maternal RSV vaccines and long-acting mAbs. 

Most HCWs (60%) recommended single-dose vaccine schedules to ensure maximum adherence and compliance. The other 40% of HCWs who recommended multidose schedules were mainly concerned with maximizing effectiveness and conferring optimal protection. Results from previous studies looking at the efficacy and completion of influenza, Human Papilloma Virus (HPV), and COVID-19 suggest that the number of doses needed to achieve optimal effectiveness can vary [20,21,22,23]. Other studies looking at dosage versus completion and coverage rates found that vaccination programs with fewer doses had higher completion and coverage rates [24,25]. These findings suggest that when designing prevention products, balancing efficacy and dosage is important.

Regarding vaccine vial sizes, most respondents preferred single-dose/device vials, mainly to avoid vaccine wastage, which aligns with findings from previous studies that found that single-dose/device vaccines substantially reduce vaccine wastage [26,27]. However, there were HCWs—especially HMOs in charge of logistics—who preferred multidose vials, with lower storage and operation costs per dose as the main reasons for their preference. Indeed, multidose vaccine vials have been previously associated with low storage and administration costs; however, previous studies evaluating vaccine wastage have indicated contamination as a major challenge with multidose vials, which leads to their recommendation of a mixed approach (single, fewer, and more dose vials) depending of the location [28,29,30]. When introducing RSV vaccines, policy makers may need to consider between single dose/device and multidose vaccine formulations, or both depending on populations and locations to match the varying needs and requirements.

Most HCWs (99%) were aware of the tetanus toxoid vaccine and recommended this vaccine to pregnant women. The majority of urban HCWs also stated that they would recommend COVID-19 vaccines for pregnant women, which was in contrast to their rural counterparts who were more likely to recommend malaria vaccines. This was not surprising as the highest prevalence of COVID-19 and malaria are reported in Nairobi and Western Kenya, respectively. Interestingly, these suggestions were made at a time when the Kenyan Ministry of Health (MoH) had not recommended COVID-19 vaccination for pregnant women, and there are no malaria vaccines available for pregnant women yet [31,32]. The recommendation of COVID-19 and malaria vaccines indicate that HCWs would recommend vaccination against diseases with proven burden, even before the vaccines become available. This further underscores the need to enhance awareness of the RSV disease burden among HCWs (and the general population) as a precursor to the introduction of new RSV prevention products.

The use of ANC clinics for the delivery of maternal vaccines, including RSV vaccines, was the most preferred strategy compared to one-time campaigns or other delivery options (including mobile outreaches). It was also noted that the main constraints towards the adequate counseling of women at ANCs were a lack of adequate training, client load and low staffing, client lateness, language barriers, and illiteracy. Some of these constraints can be addressed by creating educational materials including posters, job aids, and short adverts that can be stuck on walls or advertised through mainstream and social media. For example, a study in Benin found that the use of job aids improved maternal understanding during ANC counseling [33]. The localization of the content and staff, the use of mentor mothers, and CHVs could also address the language barrier and illiteracy constraints.

This study had one outstanding limitation. Only HCWs from two counties were interviewed, and therefore the results may not be generalizable to other counties and the country at large since healthcare has been decentralized by the 47 county governments in Kenya, and these counties may have different partners either conducting research or implementing different programs, which may cause a variation in knowledge levels. However, the HCWs are trained by national institutions, and there was no difference in RSV disease knowledge between the two counties in this analysis; therefore, minor differences in knowledge and awareness of RSV disease and RSV vaccination by counties are expected.

## 5. Conclusions

Most Kenyan HCWs were not aware of RSV disease and RSV prevention products in development. The creation of RSV awareness among the HCWs through continuous medical education, seminars, conferences, webpages, social media, and other means, including advertisements, is thus necessary. Furthermore, the findings that most HCWs preferred single-dose scheduling and single-dose vials/devices should the vaccines be available points to the balance that needs to be struck by vaccine manufactures and policymakers to consider adherence/completion rates versus efficacy, and wastage versus storage costs. This study provides important and early insights from the perspective of HCWs that can inform the successful development and implementation of RSV vaccines in Kenya and other similar low- and middle-income countries.

## Figures and Tables

**Table 1 vaccines-11-01055-t001:** Sociodemographic characteristics of the study participants, N = 106.

Characteristic	Frontline Healthcare Workers (HCWs)			Health Management Officers (HMOs) *			Total, n = 106
	Siaya	Nairobi	All	Siaya	Nairobi	All	
	n = 54	n = 40	n = 94	n = 6	n = 6	n = 12	
	n (%)	n (%)	n (%)	n (%)	n (%)	n (%)	n (%)
Gender							
Female	29 (53.7)	30 (75.0)	59 (62.8)	2 (33.3)	3 (33.3)	5 (41.7)	64 (60.4)
Job description							
Nurse	37 (68.5)	28 (70.0)	65 (69.2)	-	-	-	65 (61.3)
Clinical officer	15 (27.8)	9 (22.5)	24 (25.5)	-	-	-	24 (22.6)
Medical Doctor, GP **	1 (1.9)	2 (5.0)	3 (3.2)	-	-	-	3 (2.8)
Medical Doctor, OB/GYN ^¥^	1 (1.9)	1 (2.5)	2 (2.1)	-	-	-	2 (1.9)
Highest Education level							0
Certificate	3 (5.6)	0 (0)	3 (3.2)	0 (0)	0 (0)	0 (0)	3 (2.8)
Diploma	43 (79.6)	23 (57.5)	66 (70.2)	0 (0)	1 (16.7)	1 (8.3)	67 (63.2)
Bachelor’s degree	7 (13.0)	15 (37.5)	22 (23.4)	6 (100)	3 (50.0)	9 (75.0)	31 (29.2)
Master’s degree	1 (1.9)	2 (5.0)	3 (3.2)	0 (0)	1 (16.7)	1 (8.3)	4 (3.8)
Doctorate degree	0 (0)	0 (0)	0 (0)	0 (0)	1 (16.7)	1 (8.3)	1 (0.9)
Years of experience							
Less than 5 years	16 (29.6)	11 (27.5)	27 (28.7)	0 (0)	0 (0)	0 (0)	27 (25.5)
5 to 15 years	29 (53.7)	16 (40.0)	45 (47.9)	3 (50.0)	1 (16.7)	4 (33.3)	49 (46.2)
Over 15 years	9 (16.7)	13 (32.5)	22 (23.4)	3 (50.0)	5 (83.3)	8 (66.7)	30 (28.3)
Religion							0
Protestant	33 (61.1)	30 (75.0)	63 (67.0)	6 (100)	4 (66.7)	10 (83.3)	73 (68.9)
Catholic	19 (35.2)	10 (25.0)	29 (30.9)	0 (0)	2 (33.3)	2 (16.7)	31 (29.2)
Muslim	0 (0)	0 (0)	0 (0)	0 (0)	0 (0)	0 (0)	0 (0)
Hindu	0 (0)	0 (0)	0 (0)	0 (0)	0 (0)	0 (0)	0 (0)
Traditionalists	2 (3.7)	0 (0)	2 (2.1)	0 (0)	0 (0)	0 (0)	2 (1.9)
Have you heard about respiratory syncytial virus (RSV) ^¥¥^							
Yes	21 (38.9)	14 (35.0)	35 (37.2)	4 (66.7)	2 (50.0)	6 (60.0)	41 (39.4)

* Ten County Ministry of Health officials and 2 members of the Kenyan National Immunization Technical Advisory Group (KENITAG); ** general practitioners; ^¥^ obstetric and gynecologist; ^¥¥^ only applies to non-KENITAG HCWs.

**Table 2 vaccines-11-01055-t002:** Knowledge, attitudes, and perceptions towards RSV disease and RSV vaccination among the 41 non-KENITAG healthcare workers aware of RSV in Siaya and Nairobi Counties, Kenya.

Knowledge/Attitude	Siaya, n = 25	Nairobi, n = 16	Overall, n = 41
	n (%)	n (%)	n (%)
**What are the symptoms of RSV infection?**			
**Cough**	16 (64.0)	10 (62.5)	26 (63.4)
**Fever**	13 (52.0)	8 (50.0)	21 (51.2)
**Difficulty breathing**	12 (48.0)	9 (56.3)	21 (51.2)
**Runny nose**	6 (24.0)	3 (18.8)	9 (22.0)
**In your opinion, what groups of people are mostly affected by RSV?**			
**Infants**	4 (16.0)	4 (25.0)	8 (19.5)
**Children**	11 (44.0)	4 (25.0)	15 (36.6)
**Immunocompromised**	8 (32.0)	6 (37.5)	14 (34.1)
**Pregnant women**	8 (32.0)	8 (50.0)	16 (39.0)
**How do you rate the burden of RSV in the Country?**			
**Very high**	1 (4.0)	1 (6.3)	2 (4.9)
**High**	9 (36.0)	5 (31.3)	14 (34.1)
**Moderate**	8 (32.0)	7 (43.8)	15 (36.6)
**Low**	3 (12.0)	1 (6.3)	4 (9.8)
**Very low**	0 (0)	1 (6.3)	1 (2.4)
**Don’t know**	4 (16.0)	1 (6.3)	5 (12.2)
**Should pregnant women be vaccinated against RSV**			
**Yes**	24 (96.0)	14 (87.5)	38 (92.7)
**Are you aware of RSV prevention products**			
**Yes**	0 (0)	2 (12.5)	2 (4.9)
**Should maternal RSV vaccines be available, would you recommend the vaccines to pregnant women**			
**Yes**	25 (100)	15 (93.8)	40 (97.6)
**Do you have any reservations about giving pregnant women an RSV vaccine**			
**Yes**	3 (12.0)	4 (25.0)	7 (17.1)

**Table 3 vaccines-11-01055-t003:** Healthcare workers’ (HCWs) awareness of vaccines delivered during pregnancy in Kenya.

Characteristic	Overall, n = 106	Frontline HCWs, n = 94	Health Management Officers (HMOs), n = 12	*p*-Value
	n (%)	n (%)	n (%)	
**Aware of vaccines available for pregnant women**				
**Yes**	105 (99.0)	93 (98.9)	12 (100)	0.99
**Vaccines that HCWs are aware of**				
**Tetanus Toxoid vaccine**	104 (99.0)	92 (98.9)	12 (100)	0.99
**Diphtheria vaccine**	32 (30.5)	24 (25.8)	8 (66.7)	0.01
**Influenza vaccine**	13 (12.4)	10 (10.8)	3 (25.0)	0.16
**Pertussis vaccine**	1 (1.0)	0 (0)	1 (8.3)	0.11
**Human Papilloma Virus Vaccine**	4 (3.8)	2 (2.2)	2 (16.7)	0.06
**COVID-19**	17 (16.2)	15 (16.1)	2 (16.7)	0.99
**Other**	12 (11.4)	11 (11.8)	1 (8.3)	0.99
**Vaccines that HCWs recommend for pregnant women**				
**Tetanus Toxoid vaccine**	80 (76.2)	69 (74.2)	11 (91.7)	0.29
**COVID-19**	39 (37.1)	31 (33.3)	8 (66.7)	0.03
**Malaria**	22 (21.0)	20 (21.5)	2 (16.7)	0.99
**Reason why recommend vaccine**				
**To protect the mother**	93 (88.6)	81 (87.1)	12 (100)	0.19
**To protect the baby**	78 (74.3)	66 (71.0)	12 (100)	0.03
**Required by government**	8 (7.6)	3 (3.2)	5 (41.7)	<0.01
**Other**	24 (23.3)	18 (19.4)	6 (50.0)	0.02
**Malaria endemic zone**	9/24 (37.5)	9/18 (50.0)	0/6 (0)	
**Aware of alternative technologies such as monoclonal antibodies currently available for pregnant women or their infants**	14 (13.2)	11 (11.7)	3 (25.0)	0.20
**Perceived gestational age at which to vaccinate for optimal effectiveness**				
**<16 weeks**	43 (40.6)	36 (38.3)	7 (58.3)	0.22
**16–32 weeks**	63 (59.4)	58 (61.7)	5 (41.7)	
**With respect to vaccination schedule, which schedule would pregnant women prefer?**				
**Single dose**	79 (74.5)	74 (78.7)	7 (58.3)	0.01
**Multiple dose**	27 (25.5)	20 (21.3)	5 (41.7)	
**With respect to vaccination schedule, which schedule would you recommend for pregnant women?**				
**Single dose**	62 (58.5)	59 (62.8)	3 (25.0)	0.03
**Multiple dose**	44 (41.5)	35 (37.2)	9 (75.0)	
**Considering vaccines presented in single dose vials/ devices or multidose vials, which one would you recommend for maternal immunization? (n = 86)**				
**Single dose**	50 (58.1)	48 (61.5)	2 (25.0)	0.06
**Multiple dose**	36 (41.9)	30 (38.5)	6 (75.0)	
**Are you concerned about adverse birth outcomes following maternal immunization?**				
**Yes**	57 (54.8)	52 (55.3)	5 (41.7)	0.54
**In your opinion, which maternal immunization strategy would be the most feasible?**				
**Antenatal care clinics**	53 (50.0)	42 (44.7)	11 (91.7)	0.01
**Mass campaigns**	23 (21.7)	22 (23.4)	1 (8.3)	
**Other (mobile outreaches)**	30 (28.3)	30 (31.9)	0 (0)	
**In your opinion, which maternal immunization strategy would be the most effective?**				
**Antenatal care clinics**	53 (50.0)	44 (46.8)	9 (75.0)	0.05
**Mass campaigns**	26 (24.5)	23 (24.5)	3 (25.0)	
**Other (mobile outreaches)**	27 (25.5)	27 (28.7)	0 (0)	

## Data Availability

De-identified data may be availed through the corresponding author upon authorization by KEMRI’s Data Governance Committee.

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
