# Peer review of "Respiratory Syncytial Virus (RSV) Disease and Prevention Products: Knowledge, Attitudes, and Preferences of Kenyan Healthcare Workers in Two Counties in 2021"

_vaccines, 2023, doi:10.3390/vaccines11061055_

Round 1
Reviewer 1 Report
Little is known about the knowledge, attitudes, and perceptions of healthcare workers (HCWs) regarding Respiratory syncytial virus (RSV) disease and prevention products in Kenya. To address this gap, a mixed methods cross-sectional survey was conducted between September 16 and October 2021 in Siaya and Nairobi Counties. The survey assessed HCWs' preferences on vaccine vial sizes, scheduling, and immunization strategies, and enrolled frontline HCWs delivering services directly at maternal and child health (MCH) departments in selected health facilities and health management officers (HMOs) in-charge of vaccine distribution, policy, and management. Of the 106 respondents interviewed, 94 (88.7%) were frontline HCWs and 12 were HMOs, with only 41 (39.4%) non-KENITAG HCWs having heard about RSV disease. Majority of the participants preferred a single dose vaccine schedule (58.5%) for maximal adherence and compliance, single dose/device vaccines (58.1%) to prevent wastage and contamination, and maternal vaccination through antenatal care clinics (50%). The survey results suggest a need for increased knowledge about RSV disease and prevention among Kenyan HCWs, and the authors recommend retraining or awareness creation to prepare for when the vaccines become available.
Minor comment:
I recommend removing the participants code given in brackets after their citation (lines 169, 175 , 186, 240, 249, 251, 264, 267 and line 269) so that they cannot be recognized by the relatively simple coding. The quoting may be presented as supplementary material rather than a main text.
The article is well-written, and I recommend that it be published in its current form after acceptance of minor comment.
Author Response
We thank reviewer #1 for taking time to keenly review our manuscript and finding it fit for publication. We have deleted all the participant codes as suggested.
Reviewer 2 Report
This study aims to explore the current knowledge, attitudes, and perceptions around respiratory syncytial virus (RSV) disease and prevention products in development among the healthcare workers (HCWs) serving in various health facilities in two diverse counties in Kenya.
The abstract section should be improved: the authors missed inserting the study's aims. Moreover, "The abstract should be a total of about 200 words maximum. The abstract should be a single paragraph and should follow the style of structured abstracts but without headings". Please, revise the abstract following the authors' guidelines.
The introduction section should be functional to the study's aims: please, improve the background information. Moreover, the study's aims should be clearly written. Please, improve this section.
The material and methods section should be improved: this section should allow repetition. I strongly encourage the authors to improve this section.
The results section should be improved. The authors should clarify the groups' composition. Moreover, the differences among groups should be justified (Frontline HealthCare Workers (HCWs)=96 vs Health Management Officers (HMOs)* n=12). In this way, the statistical test is not appropriate, considering the differences in the groups' composition.
The discussion section should be improved by comparing the findings of this study with international literature: as proposed, it is redundant with the results section. Moreover, I strongly encourage the authors to improve the limitation section, inserting it as a separate section.
Finally, consider the opportunity to insert the conclusion section.
another point:
I suggest avoiding the use of the first person in a scientific paper.
Moderate editing of English language is required
Author Response
We thank reviewer #2 for the great comments that have helped us improve the manuscript. We have revised the manuscript taking the suggestions into consideration. Please also note that we have taken note of the language concern and revised appropriately except for the quoted text where we present the respondents’ verbatim. Below please find our point-by-point response.
The abstract section should be improved: the authors missed inserting the study's aims. Moreover, "The abstract should be a total of about 200 words maximum. The abstract should be a single paragraph and should follow the style of structured abstracts but without headings". Please, revise the abstract following the authors' guidelines.
Response:
Thank you for this suggestion, we have revised the abstract in line with the author’s guidelines.
The introduction section should be functional to the study's aims: please, improve the background information. Moreover, the study's aims should be clearly written. Please, improve this section.
Response:
Thank you for this observation. We have revised sections of the introduction and revised the study aims (lines 56-66).
The material and methods section should be improved: this section should allow repetition. I strongly encourage the authors to improve this section.
Response:
We have revised this section to improve its clarity. With the who, where, when, what (study questionnaire) and how explained, we strongly believe that our methods can be replicated elsewhere.
The results section should be improved. The authors should clarify the groups' composition. Moreover, the differences among groups should be justified (Frontline HealthCare Workers (HCWs)=96 vs Health Management Officers (HMOs)* n=12). In this way, the statistical test is not appropriate, considering the differences in the groups' composition.
Response:
Many thanks for this comment. The main difference between the two groups is that the HCWs provide services to pregnant women directly and are the ones who immunize them and their infants on a daily basis whereas HMOs are the county health managers in-charge of vaccine distribution and monitoring. We have stated the difference in the two groups in the methods lines 76-81 and explained the group composition in paragraph one of the results. We are indeed cognizant of the sample size and limitation of using statistical tests. We use Fisher Exact statistical test which allows us to tease out the differences in small sample sizes where necessary. In accordance with your suggestion, we sparingly use statistical tests in this work (limited to lines 183-187 only). We mainly use counts and percentages to summarize our findings.
The discussion section should be improved by comparing the findings of this study with international literature: as proposed, it is redundant with the results section. Moreover, I strongly encourage the authors to improve the limitation section, inserting it as a separate section.
Response:
We appreciate this comment. We have discussed our results and compared them to the available international literature. For the limitations, we have compared our manuscript to other articles published in the journal and recognize that the limitations are part of the discussion in most articles. We have however found this comment invaluable and structured the discussion appropriately.
Finally, consider the opportunity to insert the conclusion section.
Response:
This is well noted. We have included the conclusion as a separate section in lines 330-339.
another point:
I suggest avoiding the use of the first person in a scientific paper.
Response:
We appreciate this comment from the reviewer. We have revised the manuscript accordingly.
Reviewer 3 Report
Estimated Authors,
I've read with great interest the present cross-sectional study on KAP of Kenyan HCWs about preventive interventions against RSV.
Innovative in both its content and in the combination of qualitative and quantitative analysis, the present study is still affected by several issues: (1) the reduced number of sampled workers; (2) the limited geographic representativity of participants (2 counties); (3) the unclear design of the analyses.
Before any further re-assessment of this study, I think that Authors should extensively and specifically address this last point. More precisely:
- how were participants actually recruited and how were the interviews performed? Please explain, and also report the response rate (i.e. we have around 100 collected questionnaires, but how many were preventively distributed?)
- the data reporting is not consistent with the embedded questionnaire: please revise the design of the tables in order to follow the blueprint represented by the delivered questionnaire; more precisely, please report the items on mAb that are or certain interest when dealing with this specific topic;
- in your questionnaire, the items focused on RSV symptoms and sequelae are open ones: in the tables, only some items are reported. Please explain whether the reported items were the only ones or further content was provided in the main text.
- Authors should explain in better details in both introduction and discussion the following topics:
a) the actual indications for RSV-targeting mAb in Kenya; their actual use and availability;
b) some glimpses about the current availability of RSV vaccines (please note that recently FDA has provided some preventive authorizations to a tentative RSV vaccine)
- the qualitative assessment of the questionnaire should be detailed in the methods section.
The overall quality of the English is mostly acceptable. Some minor typos are scattered across the main text but can be solved through accurate double checking.
Author Response
We thanks Reviewer #3 for taking time to thoroughly read our manuscript and providing us with these useful comments. We have considered the suggestions and updated/improved the manuscript appropriately. Please also note that we have taken note of the language concern and revised appropriately except for the quoted text where we present the respondents’ verbatim. Please find the point by point responses below.
Innovative in both its content and in the combination of qualitative and quantitative analysis, the present study is still affected by several issues: (1) the reduced number of sampled workers; (2) the limited geographic representativity of participants (2 counties); (3) the unclear design of the analyses.
Before any further re-assessment of this study, I think that Authors should extensively and specifically address this last point. More precisely:
Response:
We appreciate the concern on small sample size and limited geography, however, we used a mixed methods (qualitative and quantitative) approach for in-depth data on the question at hand. The target population being skilled respondents working in the clinical practice and the choice of in-depth interviews using both qualitative and quantitative methods, the sample size in the current study adequately addresses the aims as a sample size of 30 is regarded as sufficient for this kind of work (https://bmcmedresmethodol.biomedcentral.com/articles/10.1186/s12874-018-0594-7). On the geographic representation – this is a limitation for this study. We have noted this in lines 321-328. Our analysis is mainly descriptive. We used counts/ frequencies and percentages/ proportions to summarize our categorical findings while medians and their IQR were used for continuous and discrete variables. We have revised the analysis section to reflect this and make it clearer.
- how were participants actually recruited and how were the interviews performed? Please explain, and also report the response rate (i.e. we have around 100 collected questionnaires, but how many were preventively distributed?)
Response:
Trained research assistants approached all HCWs assigned to the MCH clinics and the purposively selected HMOs, consented them, then conducted in-depth interviews with them. Interviews were rescheduled on respondents’ request– we therefore had a 100% response rate (included in line 118).
- the data reporting is not consistent with the embedded questionnaire: please revise the design of the tables in order to follow the blueprint represented by the delivered questionnaire; more precisely, please report the items on mAb that are or certain interest when dealing with this specific topic;
Response:
Thank you for raising this point. When collecting the data we started with the known (current vaccines and practice) to the unknown (Knowledge about RSV disease and prevention). With the unknown being our main focus, we felt that we should focus our reporting on that before reporting other findings. On the mAb suggestion, we have covered this in the text – lines 192 and 193 with addition to table 3 and Sup table 1, thank you.
- in your questionnaire, the items focused on RSV symptoms and sequelae are open ones: in the tables, only some items are reported. Please explain whether the reported items were the only ones or further content was provided in the main text.
Response:
Thank you for this comment. We had more symptoms but only selected the top four most mentioned symptoms as noted in lines 130 and 131.
- Authors should explain in better details in both introduction and discussion the following topics:
- a) the actual indications for RSV-targeting mAb in Kenya; their actual use and availability;
Response:
This is a good point. RSV-mAbs are not available in Kenya, but private facilities may import them for their at risk patients who can afford them. This is however not documented, we therefore note in line 40 that they are rarely used in Kenya and other LMICs.
- b) some glimpses about the current availability of RSV vaccines (please note that recently FDA has provided some preventive authorizations to a tentative RSV vaccine)
Response:
This is right, with many manufactures concluding their trials, the RSV vaccine field is evolving very fast. However, we have included this information in the first paragraph of the introduction. Currently the only FDA approved RSV vaccine is the GSK vaccine targeting adults >60 years.
- the qualitative assessment of the questionnaire should be detailed in the methods section.
Response:
Thank you for this observation. We believe that we have provided this information and we draw the reviewer’s attention to lines 105-108.
Round 2
Reviewer 2 Report
The authors have improved their manuscript.
English language is acceptable.
Reviewer 3 Report
The present paper has been properly improved according to my previous recommendations.
I've no further requests.
The overall quality of the English text has been radically improved.